**Data Availability Statement:** All relevant data are within the manuscript and its Supporting Information files.

**Funding:** The authors received no specific funding for this work.

# Diurnal variations of amplitude of accommodation in different age groups

**Sun-Mi Park[1], Byeong-Yeon Moon[2], Sang-Yeob Kim[2], Dong-Sik Yu[2]***

**1** Department of Optometry and Vision Science, Kyungwoon University, Gumi, Korea, **2** Department of Optometry, Kangwon National University, Samcheok, Korea

* yds@kangwon.ac.kr

## Abstract

Clinical assessment of amplitude of accommodation (AA) involves measuring the ability of the eye to change its optical power and focus on near tasks/objects. AA gradually decreases with increasing age. However, details of age-related diurnal changes in AA are not well known. This study compared diurnal changes in AA in the adolescents, the twenties, and the forties age groups. Measurement of AA using the push-up method was performed in six sessions at two-hourly intervals for 154 subjects (48, 56, 50 subjects for the adolescents, twenties, and forties age groups, respectively); the first measurements were taken from 9:00–10:00 a.m. and the final measurements from 7:00–8:00 p.m. The mean AA was 14.67 D (highest: 16.15 D in the 3:00–4:00 p.m. session, lowest: 13.35 D in the 9:00–10:00 a.m. session) for the adolescent group; 11.13 D (highest: 11.69 D in the 3:00–4:00 p.m. session; lowest: 10.61 D in the 9:00–10:00 a.m. session) in the twenties group; and 5.53 D (highest: 5.80 D in the 1:00–2:00 p.m. session, lowest: 5.11 D in the 7:00–8:00 p.m. session) in the forties age group. The measured AA showed significant difference between sessions; however, diurnal variations were greater in the younger groups. The measured AA was low at the beginning of the day in the adolescents and twenties groups and low at the end of the day in the forties age group. All age groups showed a high AA during the afternoon hours of the day (1:00–4:00 p.m.). Since the difference between each session was larger in younger subjects, AA should be evaluated while taking the age-related diurnal variations into account.

## Introduction

Accommodation is the ability of the eye to change its optical power to see objects clearly at any distance; this is done by contraction of the ciliary muscles, which are innervated by the parasympathetic nervous system [1]. Accommodative ability is at its peak in childhood and gradually declines to its lowest at about 55–60 years of age [2, 3]. Presbyopia, which is related to poor accommodative ability, presents at or before 40–45 years of age [4, 5]. Most non-presbyopic accommodative disorders are associated with the need to sustain the increased accommodation required for near work or with various drugs and certain systemic diseases such as

**Competing interests:** The authors have declared that no competing interests exist.

diabetes mellitus and myasthenia gravis [6, 7]. The prevalence of accommodative dysfunction ranges from 60 to 80% for patients with binocular vision problems [7].

A clinical assessment of accommodative function includes the measurement of amplitude of accommodation (AA), accommodative facility, and accommodative response [8]. AA, the maximum amount of accommodation that the eye can exert, is particularly important as a differential diagnosis of accommodative disorders such as accommodative insufficiency, accommodative excess, and accommodative infacility [7]. Measurement of AA, performed in preliminary examinations or binocular vison tests, is essential for the diagnosis of accommodative disorders or binocular vision disorders related to accommodation [9].

AA is measured clinically with various subjective or objective methods (push-up, push-down, minus lens-to-blur, retinoscopy, and open-field auto-refractor) [10, 11]. The accuracy of subjective measurement of AA depends on the patient's awareness of a blur point [10], the depth of focus [12], target size [13], testing conditions such as amount/intensity of illumination [10], proximal cues [14], and pupil size [15]. The push-up method among subjective methods, has the disadvantage of slightly overestimating the AA because of the relative distance magnification of the letter target, but this method is quick and easy to perform. In the Burns et al.'s review [10], they reported suggestions for improving and standardizing the clinical assessment of AA. Until recently, the majority of studies have been conducted using methodologies that had sources of error in the measurement of AA, but few studies focused on the evaluation of the change in AA in relation to the time of the day. It is common knowledge that several previous studies have proven that accommodation ability decreases with age [16–18]; age-related changes in accommodative dynamics or AA measured for a short period (ranged from seconds to minutes) have been reported in a few studies [11, 19]. In clinical practice, however, it is more important to assess AA based on the time of the day [7] or the patient's age [16]. Moreover, the assessment of AA should be a simple technique to measure. Regarding AA in relation to age, using Hofstetter's formula $(18.5 - 0.3 \times \text{age})$ can provide the expected approximate values for a given age range [16]. AA is most commonly measured using the push-up method, which is easy to perform and readily available in most clinical settings [8].

Interestingly, there is a scarcity of information on diurnal changes in AA in the available literature. Lee et al. [20] reported on diurnal changes in the accommodative functions of pre-presbyopes while they did near work. Kurtev et al. [21] and Krumholz et al. [22] also demonstrated diurnal variations in tonic accommodation in some subjects. A recent study reported that device usage has increased across all age groups, so much so that extensive daily use is now normal [23]. From the reports of previous studies, we considered that AA could be influenced by not only the subject's age, but the time of the day as well.

In this study we sought to investigate the diurnal variation of AA in three age groups (adolescents, twenties, and forties) and determine the significance of measuring AA at different times of the day, using the push-up method, which is easily performed for the clinical assessment of accommodative functions.

## Materials and methods

### Subjects

Our study was performed on only three specific age groups (adolescents, twenties, and forties). The normal age-related visual functions and characteristics of each of these three groups differ and these functions are at par with the visual demands of each age group such as learning at school with the adolescent's eye, leaving school and working at a job with the young adult's eye, and encountering problems with near vision around age forty with the presbyopic eye. For the present study, we included subjects with refractive errors free of amblyopia and within

Morgan's criteria (0–2 prism diopters (Δ) exophoria at distance; 0–6 Δ exophoria at near) as determined by the cover test [8]. The total number of participants was 154 and they were distributed into three age groups as follows: 48 subjects (mean age, 15.23 ± 3.05 years; 27 females and 21 males) for the adolescents (10s) group, aged 10 to 19 years; 56 subjects (mean age, 23.21 ± 1.93 years; 32 females and 24 males) for the twenties (20s) group, aged 20 to 29 years; and 50 subjects (mean age, 42.32 ± 2.32 years; 30 females and 20 males) for the forties (40s) group, aged 40 to 49 years. We excluded subjects with strabismus and ocular diseases such as glaucoma, cataract, and retinal disease and those with a history of prior surgery, which was determined by history-taking. The included participants had best corrected visual acuities of ≥ 20/20. Their mean refractive errors were −2.83 ± 2.06 D for spherical errors and −0.35 ± 0.53 D for cylindrical errors for the right eye, and −2.51 ± 2.07 D for spherical errors and −0.39 ± 0.58 D for cylindrical errors for the left eye. This study was approved by Kangwon National University institutional review board (KWNUIRB-2019-06-010) and written informed consent was obtained from adult participants, and from parents or guardians of minors, and the study adhered to the tenets of the Declaration of Helsinki.

## Experimental protocol

Prior to the AA measurement, all participants were refracted at a distance of 6 meters with a phoropter (CV-3000; Topcon corporation, Tokyo, Japan) and a visual chart (ACP-8; Topcon corporation, Tokyo, Japan).

The AA was measured with monocular accommodation for only the right eyes using the push-up method with an accommodative convergence rule (GR50, Bernell, USA) and a near target (near visual acuity, 20/32) under an overhead lighting (approximately 410 lux) [10, 24]. To test the right eye, the subject's left eye was occluded. The subject focused on the target placed 50 cm away and the target was brought closer to the eye at a speed of 2 cm/s [25]. The subject was instructed to keep focusing on the target and report when the target became and remained blurred. The AA was determined by measuring the distance from the point where the target became blurred to the spectacle plane and was performed with the subject's spectacle correction in place or without correction as applicable. The AA was expressed as a dioptric change between distance (0 D with full correction) and near point. This was done three times for each subject and the average was calculated [8, 26].

The AA was measured at two-hourly intervals ('midmorning', 9:00–10:00 a.m. as S1; 'late morning', 11:00–12:00 a.m. as S2; 'early afternoon', 1:00–2:00 p.m. as S3; 'midafternoon', 3:00–4:00 p.m. as S4; 'late afternoon', 5:00–6:00 p.m. as S5; 'evening', 7:00–8:00 p.m. as S6) in a total of six sessions for the three age groups. The first session took approximately 20 minutes for the completion of all measurements including preliminary examination and refraction, and the subsequent sessions for only AA were completed within five minutes. Subjects undertook their regular activities between sessions.

## Data analysis

The data generated from the sessions were collected and analyzed using IBM SPSS Statistics Version 19 (IBM Corp., USA). The three groups and six sessions were compared using a one-way analysis of variance (ANOVA) with a Bonferroni *post-hoc* correction and the one-sample t-test was used to evaluate differences between the mean and standard deviation values for the AA measured in each session. A p-value of ≤0.05 was considered significant. A receiver operating characteristic (ROC) curve analysis was performed to evaluate the discriminating ability to detect differences between the measured AA in each session.

## Results

An outline of the mean and standard deviation of the AA measured in sessions one to six (S1–S6) for the adolescents (n = 48), twenties (n = 56), and forties (n = 50) age groups is shown in Table 1. One-way ANOVA analyses showed significant differences, in descending order of values, among the three age groups for the same sessions ($p < 0.001$ for all sessions). Repeated measures ANOVA also showed significant differences among sessions for the same age groups ($p < 0.001$ for all groups; Bonferroni's *post-hoc*: S1<S2<S3<S4 and S6<S5<S4 for the adolescents age group, S1<S2<S3 and S6<S5<S4≈S3 for the twenties age group, S6<S5<S4<S3 and S1≈S2≈S3 for the forties age group). Additionally, there were significant differences between the overall mean, maximum mean, and minimum mean of the AA measured in S1–S6 for each age group. In this case, 'maximum and minimum mean' represents an average of the highest and lowest measured AA for each subject in each session as outlined in Table 1. These findings are also shown in Fig 1 in comparison to the calculated AA derived with Hofstetter's equations. The overall mean and the calculated AA were 14.67 D and 13.93 D for the adolescents age group, 11.13 D and 11.54 for the twenties age group, and 5.53 D and 5.80 D for the forties age group; there was little difference between the values of the overall mean AA and calculated AA.

The comparisons of the mean AA measured in each session and the overall mean AA for each age group are shown in Table 2. For the adolescents age group, the AA value in S4 was higher than ($p = 0.010$) and the AA in S1 lower than overall mean ($p = 0.005$), whereas for the forties age group, the AA value in S6 lower than overall mean ($p = 0.008$). For the twenties age group, there were no significant differences between the mean AA measured in all sessions

**Table 1. Mean and standard deviation (SD) for the AA measured in each of the six sessions.**

| Session | Adolescents (n = 48) | Twenties (n = 56) | Forties (n = 50) | One-way ANOVA[‡] |
|---|---|---|---|---|
| S1 | 13.35 ± 3.13 | 10.61 ± 2.75 | 5.55 ± 1.28 | $F_{(2, 151)} = 121.99$, $p < 0.001$ (a>b>c) |
| S2 | 13.98 ± 3.24 | 10.88 ± 2.86 | 5.68 ± 1.19 | $F_{(2, 151)} = 129.16$, $p < 0.001$ (a>b>c) |
| S3 | 15.42 ± 3.71 | 11.43 ± 2.74 | 5.80 ± 1.12 | $F_{(2, 151)} = 154.61$, $p < 0.001$ (a>b>c) |
| S4 | 16.15 ± 3.80 | 11.69 ± 3.09 | 5.68 ± 1.15 | $F_{(2, 151)} = 161.44$, $p < 0.001$ (a>b>c) |
| S5 | 14.97 ± 3.34 | 11.24 ± 2.79 | 5.35 ± 1.14 | $F_{(2, 151)} = 172.53$, $p < 0.001$ (a>b>c) |
| S6 | 14.14 ± 3.14 | 10.94 ± 2.72 | 5.11 ± 1.09 | $F_{(2, 151)} = 167.86$, $p < 0.001$ (a>b>c) |
| Repeated measures ANOVA[‡] | $F_{(2.97, 139.35)} = 55.81$, $p < 0.001$ S1<S2<S3<S4, S6<S5<S4 | $F_{(3.29, 181.06)} = 15.52$, $p < 0.001$ S1<S2<S3, S6<S5<S4(≈S3) | $F_{(1.98, 96.95)} = 19.98$, $p < 0.001$ S6<S5<S4<S3, S1≈S2≈S3 | |
| S1–S6[†] | | | | |
| Overall mean | 14.67 ± 3.29 | 11.13 ± 2.74 | 5.53 ± 1.10 | $F_{(2, 151)} = 160.85$, $p < 0.001$ (a>b>c) |
| Maximum mean | 16.19 ± 3.83 | 12.07 ± 2.94 | 6.04 ± 1.15 | $F_{(2, 151)} = 157.37$, $p < 0.001$ (a>b>c) |
| Minimum mean | 13.18 ± 2.93 | 10.26 ± 2.70 | 5.01 ± 1.13 | $F_{(2, 151)} = 147.05$, $p < 0.001$ (a>b>c) |

Unit: diopter (D);

[†]Values obtained for the overall, maximum, and minimum mean AA in the six sessions for each age group.

[‡]Bonferroni's *post-hoc*; a: adolescents; b: twenties; c: forties. S1: 9:00–10:00 a.m.; S2: 11:00–12:00 a.m.; S3: 1:00–2:00 p.m.; S4: 3:00–4:00 p.m.; S5: 5:00–6:00 p.m.; S6: 7:00–8:00 p.m. AA: amplitude of accommodation, ANOVA: analysis of variance.

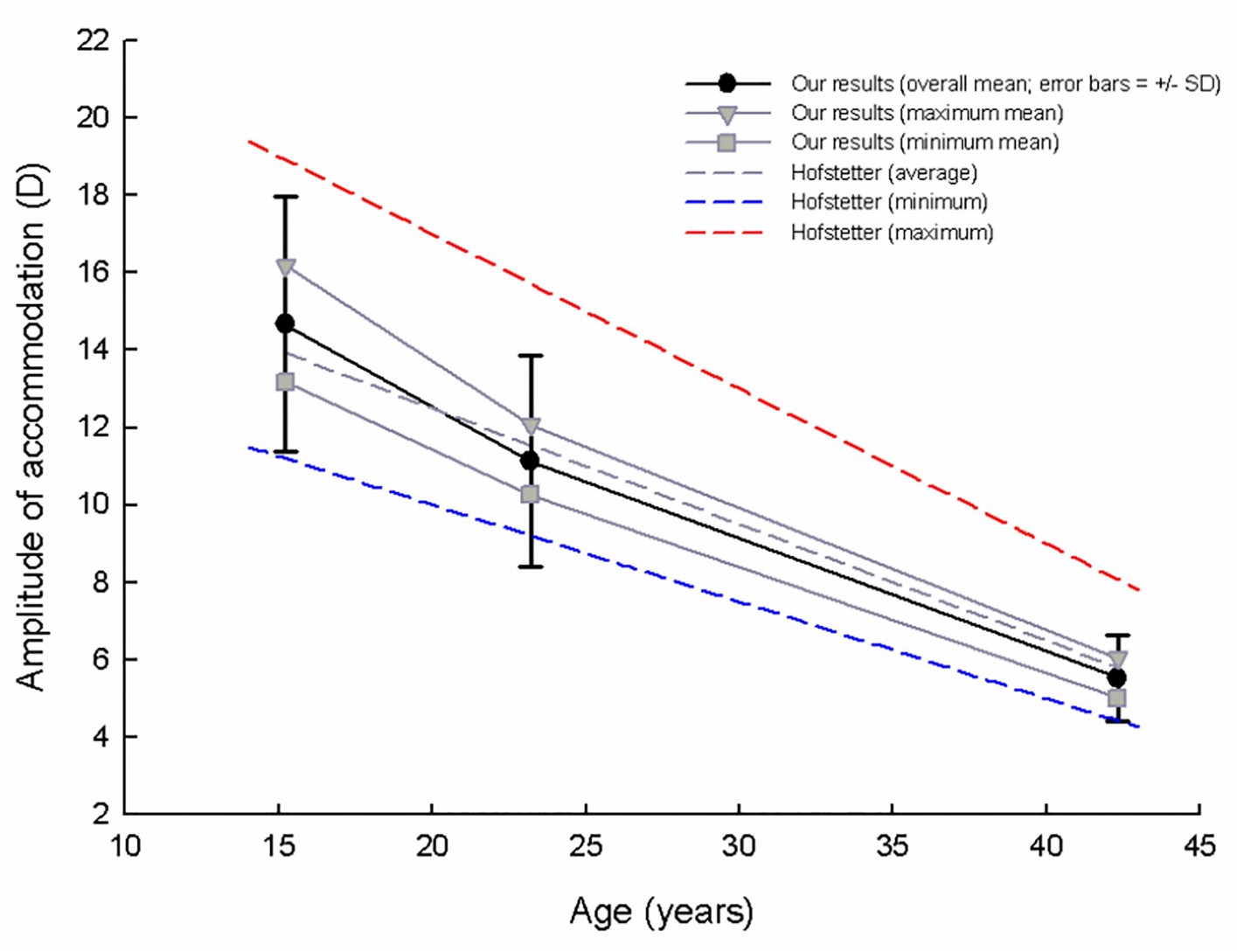

**Fig 1. Comparison of our results and calculated AA from Hofstetter's equation.**

and overall mean. Compared to S1 as a baseline, diurnal variations in AA were 0.80 D–2.80 D for the adolescents age group, 0.27 D–1.08 D for the twenties age group, 0.13 D–0.44 D for the forties age group as shown in Fig 2. Mean changes in AA were 1.32 D for the adolescents age group, 0.52 D for the twenties age group and −0.02 D for the forties age group.

**Table 2. Analysis of the difference between the AA in each session and the overall mean using the one-sample t-test.**

| | | Adolescents (n = 48) | Twenties (n = 56) | Forties (n = 50) |
|---|---|---|---|---|
| Overall mean | | 14.67 | 11.13 | 5.53 |
| One-sample t-test | S1 | t(47) = 2.93, p = 0.005 | t(55) = 1.42, p = 0.160 | t(49) = 0.09, p = 0.927 |
| | S2 | t(47) = 1.48, p = 0.146 | t(55) = 0.66, p = 0.509 | t(49) = 0.91, p = 0.368 |
| | S3 | t(47) = 1.40, p = 0.168 | t(55) = 0.82, p = 0.414 | t(49) = 1.70, p = 0.096 |
| | S4 | t(47) = 2.70, p = 0.010 | t(55) = 1.34, p = 0.185 | t(49) = 0.92, p = 0.363 |
| | S5 | t(47) = 0.62, p = 0.539 | t(55) = 0.30, p = 0.765 | t(49) = 1.15, p = 0.255 |
| | S6 | t(47) = 1.16, p = 0.250 | t(55) = 0.52, p = 0.602 | t(49) = 2.75, p = 0.008 |

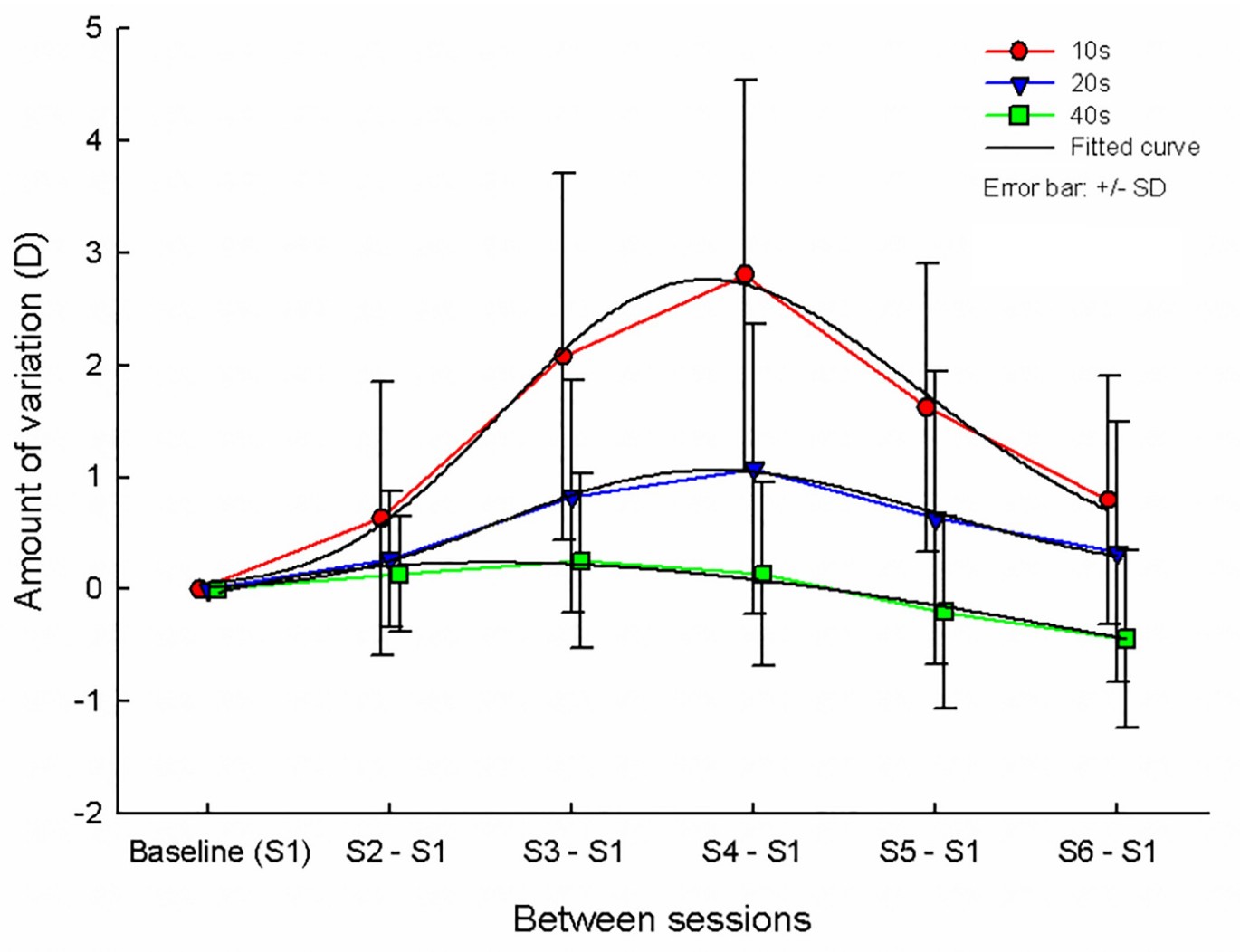

**Fig 2. Diurnal variation in amplitude of accommodation.**

For each session, the measured AA for each participant was compared to the mean AA to determine if the AA has increased or decreased; the results of this analysis are shown in Fig 3. Values for each session are presented in ascending order regardless of the testing order of the participants. In the adolescents group, the number of increases versus decreases was 1:47 for S1, 5:43 for S2, 39:9 for S3, 47:1 for S4, 34:14 for S5, and 17:3 for S6. In the twenties age group, zero values were excluded from the number of changes in AA, and the number of increases versus decreases was 10:44 for S1, 19:35 for S2, 37:17 for S3, 43:11 for S4, 29:25 for S5, and 17:37 for S6. In the forties age group, zero values were also excluded, and the number of increases versus decreases was 20:28 for S1, 31:17 for S2, 42:6 for S3, 37:11 for S4, 13:34 for S5, and 3:45 for S6.

We assessed the discriminative ability of each session for detecting differences using ROC curve analysis [27]. The ROC curve analysis gives a statistical test comparing the area under the curve (AUC) to the value 0.5 for each session. The small p-values ($<0.05$) indicate a significant difference from 0.5 for each session. The AUC analysis also gives a 95% confidence interval for each estimated AUC. The results of the ROC analysis are shown in Table 3. Regarding

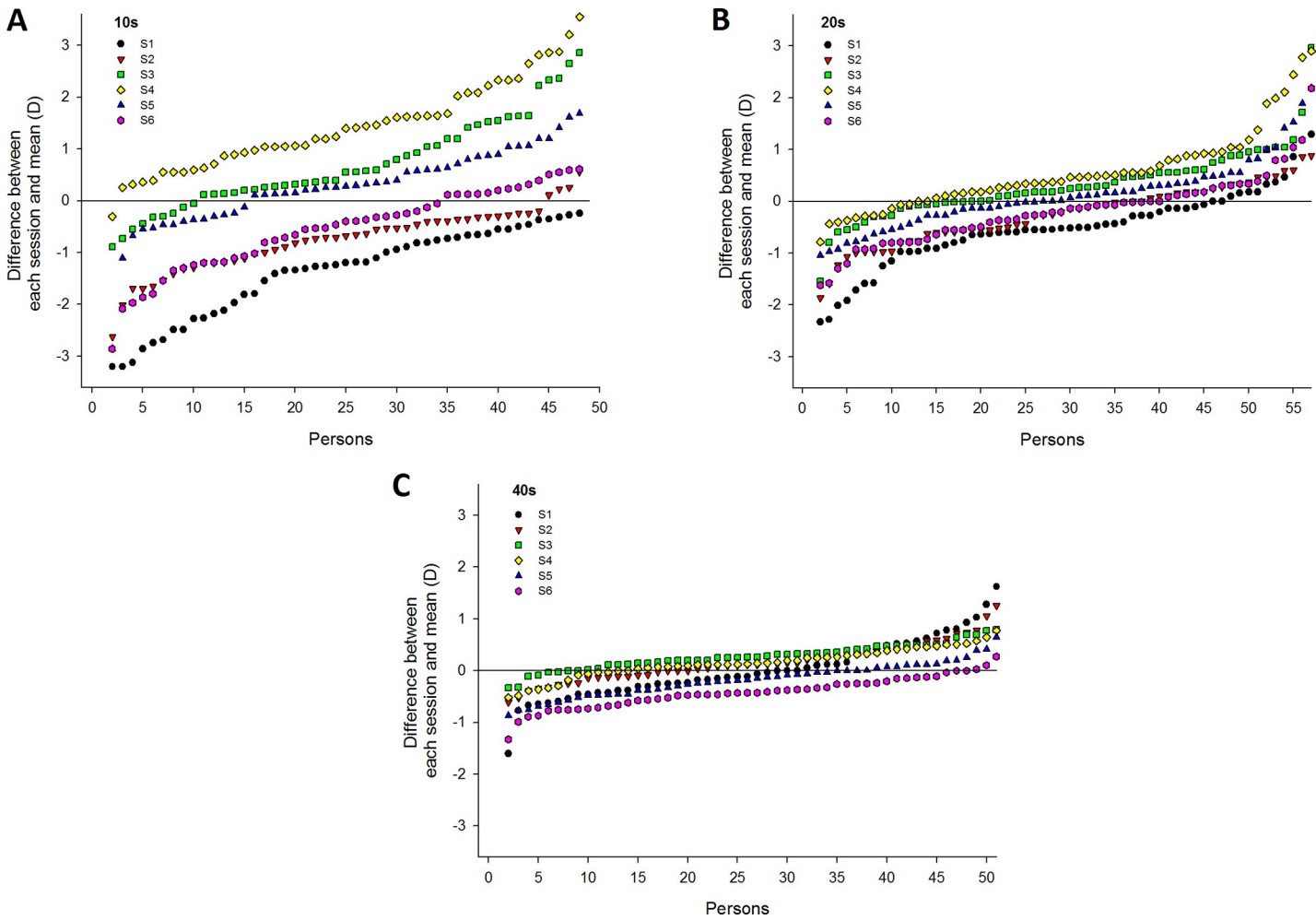

**Fig 3. Differences between the mean AA and the AA measured in each session for each participant.** (A) Adolescents group. (B) Twenties group. (C) Forties group. The data are sorted in ascending order regardless of the testing order of the participants.

detection of differences ($>1.50$ D) among AA values measured in each session, the AUC was least (0.795) for S1 and greatest (0.833) for S4 in all age groups, and least (0.792) for S1 and greatest (0.919) for S4 in adolescents. However, there was no significant AUC in the twenties and forties age groups. In comparison of the ROC curves between sessions, there were differences between sessions in the adolescents group but there was no difference between the sessions in the twenties and the forties group. The significant difference between sessions in all age groups may be due to the influence of twenties age group.

Fig 4 shows ROC curves indicating that S4 had high discrimination ability for the adolescents and twenties groups whereas S2 had a high discrimination ability for the forties group. S4 for the adolescents group showed a discriminative ability for detecting differences by the ROC curve analysis. The AA in S4 was high in the adolescents group compared to other sessions. S4 in the twenties group and S2 in the forties group had no discriminative difference in AA compared to other sessions.

Regarding detection of differences ($>0.75$ D) among AA values measured at the sessions, the area under the curve (AUC) was 0.762 (95% CI, 0.522–1.000; p = 0.026) for S6 to 0.792 (95% CI, 0.557–1.000; p = 0.013) for S4 in twenties age group. However, there was no significant AUC in the forties age group.

**Table 3. ROC analysis for detecting differences (>1.50 D) among the measured AA in each session.**

| Session | AUC (95% CI), p-value | | | |
|---|---|---|---|---|
| | All ages (N = 154) | Adolescents (n = 48) | Twenties (n = 56) | Forties (n = 50) |
| S1 | 0.795 (0.725–0.866), p < 0.001 | 0.792 (0.642–0.942), p = 0.007 | 0.560 (0.407–0.713), p = 0.441 | 0.661 (0.457–0.865), p = 0.153 |
| S2 | 0.806 (0.738–0.874), p < 0.001 | 0.842 (0.720–0.964), p = 0.002 | 0.566 (0.413–0.719), p = 0.398 | 0.674 (0.501–0.847), p = 0.122 |
| S3 | 0.822 (0.757–0.887), p < 0.001 | 0.905 (0.820–0.989), p < 0.001 | 0.595 (0.441–0.748), p = 0.225 | 0.612 (0.459–0.764), p = 0.321 |
| S4 | 0.833 (0.770–0.896), p < 0.001 | 0.919 (0.839–0.998), p < 0.001 | 0.618 (0.468–0.768), p = 0.129 | 0.564 (0.401–0.727), p = 0.569 |
| S5 | 0.826 (0.762–0.890), p < 0.001 | 0.872 (0.760–0.984), p = 0.001 | 0.596 (0.446–0.747), p = 0.216 | 0.589 (0.425–0.754), p = 0.427 |
| S6 | 0.816 (0.749–0.883), p < 0.001 | 0.821 (0.665–0.976), p = 0.003 | 0.609 (0.460–0.758), p = 0.161 | 0.521 (0.351–0.691), p = 0.853 |
| Comparison of ROC curve between sessions (p) | S1 : S3 (p = 0.033)<br>S1 : S4 (p = 0.003)<br>S1 : S5 (p = 0.018)<br>S2 : S4 (p = 0.004)<br>Others (p > 0.05) | S1 : S3 (p = 0.005)<br>S1 : S4 (p = 0.002)<br>S1 : S5 (p = 0.016)<br>S2 : S3 (p = 0.019)<br>S2 : S4 (p = 0.006)<br>S3 : S6 (p = 0.042)<br>S4 : S6 (p = 0.010)<br>S5 : S6 (p = 0.042)<br>Others (p > 0.05) | All pairs (p > 0.05) | All pairs (p > 0.05) |

ROC: receiver operating characteristics; AUC: area under the curve; CI: confidence interval

## Discussion

The assessment of accommodation is one of the most important tests of visual function. The AA is an accommodative function that is conventionally measured with a simple and quick push-up method in clinical practice [28, 29]. AA measured with the push-up method may be considerably influenced by ocular parameters [12, 15], measurement conditions [10, 13], and psychological factors [10, 14] affecting the patient. However, diurnal variations in AA measured with the push-up method have not been reported. The main goal of this study was to evaluate the diurnal variations in the AA measured for different age groups. As observed in our results, the diurnal variations in AA between sessions showed significant differences and

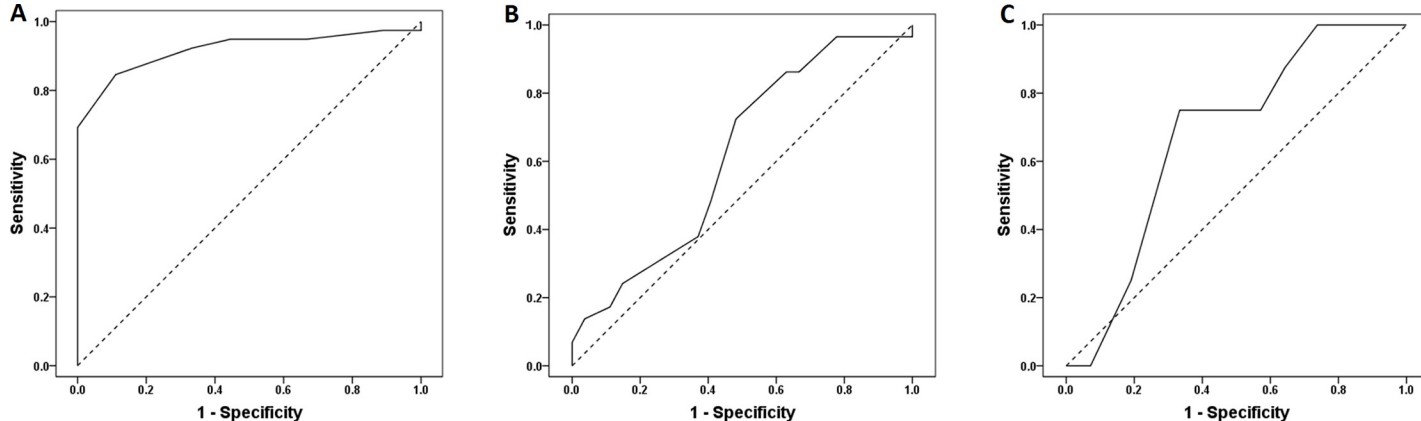

**Fig 4. ROC curve analysis showing sessions with high discrimination ability for each age group.** (A) S4 in the adolescents group. (B) S4 in the twenties group. (C) S2 in the forties group. Dashed diagonal lines are reference lines, which has an AUC of 0.5.

the extent of the variation was clinically meaningful; notably, diurnal variations were greater in the younger age groups.

In the comparative analysis of all measurement sessions for each age group, all age groups showed a significant difference in AA (Table 1). The difference between the greatest and the smallest AA for all sessions was 2.80 D for the adolescents group, 1.08 D for the twenties group, and 0.69 D for the forties group, indicating that the difference reduces with increasing age. These differences can affect the assessment of accommodative functions, especially in the case of an AA placed in a "borderline accommodative insufficiency" category [30] when the diagnostic criterion for accommodative insufficiency is 2.00 D below the mean AA derived from Hofstetter's equation [31]. Therefore, the evaluation of accommodation in younger age groups may be misinterpreted, depending on time of measurement, if the value of the measured AA is close to the diagnostic criterion for accommodative insufficiency.

A previous study of diurnal change in AA in adults (18 to 19 years) had indicated that AA was reduced in the evening compared to the morning [32]. In this previous report, the study only presented the frequency of AA measured with the push-up method in the morning and evening (the measuring time was unidentified) and did not provide a detailed analysis of each session. In our study, sessions with the greatest and the smallest AA were S4 (3:00–4:00 p.m.) and S1 (9:00–10:00 a.m.) for the adolescents and twenties groups respectively, and S3 (1:00–2:00 p.m.) and S6 (7:00–8:00 p.m.) for the forties group. These results indicate that the AA in younger age groups decreases from its peak during the afternoon hours of 1:00–4:00 p.m.; lower AA is more likely to be recorded during the midmorning hours in younger age groups and in the evening in older age groups.

In the comparative analysis of maximum and minimum AA recorded in each measurement session, the differences in AA were significant in all age groups but the differences were greater in the younger age groups (Table 1). The difference between the maximum and minimum mean AA measured in each session for each participant was 3.10 D for the adolescents group, 1.81 D for the twenties group, and 1.03 D for the forties group, with greater difference recorded in younger age groups. Linear regressions derived from overall mean, maximum and minimum mean AA for each session was $19.31 − (0.33 \times age)$, $R^2 = 0.983$; $21.23 − (0.36 \times age)$, $R^2 = 0.985$; $17.47 − (0.30 \times age)$, $= 0.995$, respectively (Fig 1). The high values of the coefficient of determination ($R^2$) clearly explain the age-related changes in AA. Each AA and the ranges between means were significantly reduced in older age groups. Linear regression for overall mean AA decreased with age, with trends similar to Hofstetter's equation $= 18.5 − (0.3 \times age)$ for average AA. Linear regressions for maximum and minimum mean AA fell in the middle of Hofstetter's equation $= 25 − (0.4 \times age)$ for maximum AA and $15 − (0.25 \times age)$ for minimum AA. Although the mean values of AA among various age groups in several studies were different from the predicted AA using Hofstetter's equations [33–35], the value of the overall mean AA in our findings was similar to the AA derived Hofstetter's equations.

In the analysis of the difference between the overall mean and mean measured AA in each session, significant differences showed a tendency to appear in sessions with low or high AA. Significant differences between the overall mean and mean AA in each session were found in the adolescents age group for S1, which had a lower AA, and S4 which had a higher AA; for the forties group, a significant difference was recorded in S6, which had a lower AA, but no significant difference was found in the twenties group (Table 2). The results of this study indicate that AA is more likely to be higher or lower than the mean AA in the midmorning and midafternoon hours in the adolescents group and in the evening in older groups. These changes may be in line with the results of previous studies that reported diurnal variations in accommodative functions and tonic accommodation, decreased AA with increased age [36] and maximum tonic accommodation or AA in the late afternoon and morning hours [21, 32].

In the evaluation of net changes between each session based on the first session, the change in AA showed a tendency to decrease after increasing, and the range of change was greater in younger age groups. The net change in AA was typically found to be lowest in the midmorning (S1) and gradually increased until midafternoon (S4) and then decreased to a higher value in the evening (S6) than that measured in the morning in the adolescents and twenties age groups. The net change in AA increased from the midmorning (S1) until the early afternoon (S3) and then decreased to a lower value in the evening (S6) than that measured in the morning in forties age group (Fig 2). The pattern of our results is similar to that of a previous research on diurnal change in AA, which indicates that AA was high in the afternoon and low in the morning in the 38–49 years age group with near works for more than seven hours a day [20]. The patterns of the changes recorded in our study were right-skewed unimodal curves for the adolescents and twenties groups or left-skewed unimodal curves for the forties age group. Therefore, the AA should be evaluated considering sessions in which the measured AA was too high or too low.

In the analysis of frequency of variation between the mean AA and the AA measured in each session, the frequency of an increase was high for S3, S4, and S5, and the frequency of a decrease was high for S1, S2, and S6 in the adolescents and twenties groups. In the forties group, the frequency of an increase was high for S2, S3, and S4, and the frequency of a decrease was high for S1, S5, and S6. The difference between measured AA for each session and the mean AA was lower in older groups than in younger groups (Fig 3).

Finally, only S4 for the adolescents group exhibited an ability to distinguish the differences between AAs for each session in the ROC curve analysis. The criterion for assessing the difference in AA was based on 1.50 D, which is greater than the 1.42 D suggested in the repeatability of clinical measurements of AA using the push-up method [37]. Discriminative ability was observed only in S4 for the adolescents group, which appeared to be distinct and exhibited large diurnal variation. Conversely, discriminative ability was non-existent in other age groups, which showed diurnal variation, even at the lower criterion of 0.75 D difference.

The limitation of this study is that the conditions of the eyes such as eye fatigue [38] and the degree of refractive error [39, 40] were not considered before AA measurements. However, these limitations in our study may not have had much impact on the results as it was possible to evaluate the tendency of change by comparing the diurnal change with the average AA measured in a normal living environment.

In summary, significant diurnal changes in AA were observed between sessions measured at two-hourly intervals from 9:00 a.m. to 8:00 p.m. and the changes were greater in younger age groups. The highest AA was measured in the midafternoon in the adolescents and twenties age groups and in the early afternoon in the forties age group. The lowest AA was measured in the midmorning in the adolescents and twenties age groups and in the evening in the forties age group. Therefore, AA should be evaluated taking into account the patient's age and time of measurement, since the lower the age, the greater the differences in the AA between the measurement sessions.

## Supporting information

**S1 File. All relevant raw data.**
(XLSX)

## Author Contributions

**Conceptualization:** Sang-Yeob Kim, Dong-Sik Yu.

**Data curation:** Sun-Mi Park, Dong-Sik Yu.

**Formal analysis:** Sun-Mi Park, Byeong-Yeon Moon, Dong-Sik Yu.

**Methodology:** Sun-Mi Park, Sang-Yeob Kim, Dong-Sik Yu.

**Project administration:** Dong-Sik Yu.

**Resources:** Sun-Mi Park.

**Supervision:** Dong-Sik Yu.

**Validation:** Byeong-Yeon Moon, Sang-Yeob Kim, Dong-Sik Yu.

**Visualization:** Byeong-Yeon Moon, Dong-Sik Yu.

**Writing – original draft:** Sun-Mi Park, Dong-Sik Yu.

**Writing – review & editing:** Sun-Mi Park, Byeong-Yeon Moon, Sang-Yeob Kim, Dong-Sik Yu.

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
