## [Decision Letter · Decision Letter 0]

24 Sep 2019

PONE-D-19-24679

Diurnal variations of amplitude of accommodation in different age groups

PLOS ONE

Dear Mr. Yu,

Thank you for submitting your manuscript to PLOS ONE. After careful consideration, we feel that it has merit but does not fully meet PLOS ONE’s publication criteria as it currently stands. Therefore, we invite you to submit a revised version of the manuscript that addresses the points raised during the review process.

We would appreciate receiving your revised manuscript by Nov 08 2019 11:59PM. To enhance the reproducibility of your results, we recommend that if applicable you deposit your laboratory protocols in protocols.io, where a protocol can be assigned its own identifier (DOI) such that it can be cited independently in the future. For instructions see: http://journals.plos.org/plosone/s/submission-guidelines#loc-laboratory-protocols

We look forward to receiving your revised manuscript.

Kind regards,

Ireneusz Grulkowski, PhD

Academic Editor

PLOS ONE

Journal Requirements:

1. Please provide additional details regarding participant consent. In the Methods section, please ensure that you have specified (1) whether consent was informed and (2) what type you obtained (for instance, written or verbal). Please state in your methods section whether you obtained consent from parents or guardians of the minors included in the study or whether the research ethics committee or IRB approved the lack of parent or guardian consent.

Additional Editor Comments (if provided):

This study involves psychology/behavioral research conducted using the online platform Amazon Mechanical Turk. Because workers on these platforms are often experienced in participating in such studies or may be motivated to participate when they should not, please confirm that you sufficiently addressed the issues of non-naivety and trustworthiness.

Reviewers' comments:

Reviewer's Responses to Questions

**Comments to the Author**

1. Is the manuscript technically sound, and do the data support the conclusions?

Reviewer #1: No

Reviewer #2: Yes

2. Has the statistical analysis been performed appropriately and rigorously? 

Reviewer #1: No

Reviewer #2: Yes

3. Have the authors made all data underlying the findings in their manuscript fully available?

Reviewer #1: Yes

Reviewer #2: No

4. Is the manuscript presented in an intelligible fashion and written in standard English?

Reviewer #1: Yes

Reviewer #2: Yes

5. Review Comments to the Author

Reviewer #1: INTRODUCTION

- Page 3 line 53-55: Why AA is important to diagnosis either accommodative insufficiency or the accommodative excess?

- Page 3 line 59-64: Why knowing all the limitations about subjective methods to measure the AA, you chosen the Push up.

- Page 4 line 71-72: “In clinical practice, however, it is more important to assess AA based on the time of the day or the patient’s age.” Why or how you support this affirmation, indeed, in real world few times a subject has to do a lot of accommodation.

- Page 4, lines 78-85: According to that paragraph, I think, it would be better to measure the accommodative response (lag) or the accommodative facility.

- I cannot see any cause that support to perform that study.

- why you considered that age could be an important variability factor in the daily changes of the AA?

RESULTS:

- Page 9, lines 201-207: I could not understand which increases or decreases you say.

- Figure 3: It is blur and I could not see it.

- Page 10-11: The analysis of Table 3 is not clear, you say, "Regarding detection of differences (> 1.50 D) among AA values measured in each session" i.e., within the three times that you measured the AA in each S1, S2 etc, or the differences between S1, S2 etc; this is important, because in the first case, you are showing that the test used is not reliable (high AUC for adolescent and twenties) and the changes in the AA could be because a high variability by the Push up test.

DISCUSSION

- Page 16, lines 352-360: “significant diurnal changes in AA were observed between sessions measured at two-hour intervals from 9:00 a.m. to 8:00 p.m. and the changes were greater in younger age groups” I do not agree with this conclusion, because the large variability within each session could cause that differences, but no a really change in the amplitude of accommodation.

Reviewer #2: As stated by the authors in the introduction section, pupil diameter is important in the accommodation amplitude measurement.Have pupil diameters of the participants been measured? If yes, I suggest you give us some information.

I also wondered if the results were different between male and female participants. The gender of the participants is not present in the Excel file.In general, I must say that I like the article.

6. PLOS authors have the option to publish the peer review history of their article (what does this mean?). If published, this will include your full peer review and any attached files.

Reviewer #1: No

Reviewer #2: No

---

## [Author Response · Author response to Decision Letter 0]

21 Oct 2019

Dear reviewers:

We thank you and the reviewers for reviewing our manuscript and the valuable comments provided. We have revised our manuscript accordingly and provided the point-by-point responses to the reviewers’ comments. Enclosed, please find the revised manuscript.All the authors express our appreciation for your kind consideration of our manuscript. 

---

## [Decision Letter · Decision Letter 1]

13 Nov 2019

Diurnal variations of amplitude of accommodation in different age groups

PONE-D-19-24679R1

Dear Dr. Yu,

We are pleased to inform you that your manuscript has been judged scientifically suitable for publication and will be formally accepted for publication once it complies with all outstanding technical requirements.

With kind regards,

Ireneusz Grulkowski, PhD

Academic Editor

PLOS ONE

Additional Editor Comments (optional):

Reviewers' comments:

Reviewer's Responses to Questions

**Comments to the Author**

1. If the authors have adequately addressed your comments raised in a previous round of review and you feel that this manuscript is now acceptable for publication, you may indicate that here to bypass the “Comments to the Author” section, enter your conflict of interest statement in the “Confidential to Editor” section, and submit your "Accept" recommendation.

Reviewer #2: All comments have been addressed

2. Is the manuscript technically sound, and do the data support the conclusions?

Reviewer #2: Yes

3. Has the statistical analysis been performed appropriately and rigorously? 

Reviewer #2: Yes

4. Have the authors made all data underlying the findings in their manuscript fully available?

Reviewer #2: Yes

5. Is the manuscript presented in an intelligible fashion and written in standard English?

Reviewer #2: Yes

6. Review Comments to the Author

Reviewer #2: thank you for the answers. I think your answers to me and the other judge are satisfactory

I think your article contains information that will contribute to the literature

congratulations

7. PLOS authors have the option to publish the peer review history of their article (what does this mean?). If published, this will include your full peer review and any attached files.

Reviewer #2: No

---

## [Editor Report · Acceptance letter]

18 Nov 2019

PONE-D-19-24679R1 

Diurnal variations of amplitude of accommodation in different age groups 

Dear Dr. Yu:

I am pleased to inform you that your manuscript has been deemed suitable for publication in PLOS ONE. Congratulations! Your manuscript is now with our production department. 

With kind regards,

on behalf of

Dr. Ireneusz Grulkowski 

Academic Editor

PLOS ONE